# Anticipated impacts of Brexit scenarios on UK food prices and implications for policies on poverty and health: a structured expert judgement approach

Martine Jayne Barons [1], Willy Aspinall[2]

[1]Department of Statistics, University of Warwick, Coventry, UK
[2]Scool of Earth Sciences, University of Bristol, Bristol, UK

**Correspondence to**
Dr Martine Jayne Barons;
Martine.Barons@warwick.ac.uk

## ABSTRACT

**Introduction** Food insecurity is associated with increased risk for several health conditions and with poor chronic disease management. Key determinants for household food insecurity are income and food costs. Whereas short-term household incomes are likely to remain static, increased food prices would be a significant driver of food insecurity.

**Objectives** To investigate food price drivers for household food security and its health consequences in the UK under scenarios of Deal and No-deal for Britain's exit from the European Union. To estimate the 5% and 95% quantiles of the projected price distributions.

**Design** Structured expert judgement elicitation, a well-established method for quantifying uncertainty, using experts. In July 2018, each expert estimated the median, 5% and 95% quantiles of changes in price for 10 food categories under Brexit Deal and No-deal to June 2020 assuming Brexit had taken place on 29 March 2019. These were aggregated based on the accuracy and informativeness of the experts on calibration questions.

**Participants** Ten specialists with expertise in food procurement, retail, agriculture, economics, statistics and household food security.

**Results** When combined in proportions used to calculate Consumer Price Index food basket costs, median food price change for Brexit with a Deal is expected to be +6.1% (90% credible interval −3% to +17%) and with No-deal +22.5% (90% credible interval +1% to +52%).

**Conclusions** The number of households experiencing food insecurity and its severity is likely to increase because of expected sizeable increases in median food prices after Brexit. Higher increases are more likely than lower rises and towards the upper limits, these would entail severe impacts. Research showing a low food budget leads to increasingly poor diet suggests that demand for health services in both the short and longer terms is likely to increase due to the effects of food insecurity on the incidence and management of diet-sensitive conditions.

## INTRODUCTION

Food insecurity, the lack of access to sufficient nutritious food, is associated with multiple negative outcomes including diet-sensitive chronic diseases. An important driver of

### Strengths and limitations of this study

► First study to quantify anticipated food price changes and associated uncertainty relating to Brexit Deal and No-deal scenarios using a transparent and established protocol, and to articulate links to potential healthcare impacts.
► Inclusion of experts with broad and overlapping areas of expertise in food production and supply.
► Scenario analysis showing how price changes in linked food categories could combine to affect overall food costs.
► Fewer than optimal number of experts elicited.
► Study was undertaken on the assumption that Brexit would occur on 29 March 2019. Delays and major ambiguities with respect to eventual Brexit circumstances have emerged since experts made projections in 2018.

household food security is the costs of food and other essentials relative to incomes. In 2016, the UK voted to relinquish its membership of the European Union (EU), known colloquially as 'Brexit', to be completed by 29 March 2019. UK reliance on food imports, including from EU, is significant and food price rises have been widely forecast.[1]

One of the motivations for investigating possible impacts of Brexit on food prices is that, over the last few years, medical general practitioners in the UK have raised concerns about food insecure patients seeking referrals to food banks[2] and that food insecurity is affecting medication compliance, health and well-being.[3] This has raised concern about resource implications for surgeries[4] and that 'the welfare system is failing to provide a robust last line of defence against hunger'.[5]

A 2017 survey showed 13% of people were worried that their food would run out before they got money to buy more ('marginally food secure households') and 8% could not afford to eat balanced meals or went hungry ('low or very low food secure households').[6] In

low-income households, 29% experience food insecurity.[7] Lower income households inevitably allocate a higher proportion of spending on food, and buy a similar fraction of imported food; therefore, low-income households are more exposed to food price rises. In November 2018, the United Nations Office of the High Commissioner for Human Rights special rapporteur issued a statement in which he argued that the rise in food bank use in the UK is a consequence of poverty, including in-work poverty. He recommended that the UK government should begin to measure and monitor food security.[8]

Absolute income levels and volatility are both important drivers of household food insecurity.[9 10] There was little growth in real earnings in 2017–2018, and the Office for Budget Responsibility forecast slow earnings growth for the following 4 years.[11] This means that the main driver of household food insecurity will be food price. UK price inflation is measured by changes in the UK Consumer Price Index (CPI). The CPI is based on a 'shopping basket' of goods and services, including a food element. In November 2018, the CPI inflation was 2.3% per annum over all items and 0.5% p.a. for the food element.[12] The CPI is based on actual consumer spending and the food element incorporates both spending on healthy nutrition and on less healthy options. CPI alone cannot indicate if consumers are shifting towards less healthy diets because confounding effects are smoothed out across income ranges.

Consequences of food insecurity on diet-sensitive chronic diseases, including hypertension, hyperlipidaemia and diabetes,[13–19] are significant. Other effects include poor educational attainment, poor mental health and social isolation, which increase mortality.[20] People on lower incomes report shopping for cheaper foods and eating less,[6] eating more high-fat, salty, sugar-sweetened foods and processed meat.[21] Fruit and vegetable consumption is lower in low-income households.[22] The medical importance of a basic nutritional safety net has long been recognised by policymakers through the Welfare Food Scheme and Healthy Start programmes. A recent systematic review identifies poor diet quality as an important preventable risk factor for non-communicable disease, responsible for one-in-five deaths globally and 127 deaths per 100 000 Britons.[23] Under post-2008 austerity measures, cuts have been made to Healthy Start and other provisions. The UK's main response to growing food insecurity has been charitable food relief, but the efficacy of these and similar approaches is unmeasured. Social protection spending and welfare state interventions are the only actions known to alter the prevalence of household food insecurity.[24] The rise in food bank use is attributed largely to welfare cuts.[4] Despite the purported end of austerity, the inability of some households to feed themselves adequately persists. Figures from The Trussell Trust, the UK's largest network of food banks, show 1 583 668 three-day emergency food supplies were issued in the 12 months to 31 March 2019, an increase of 19% over the same period in 2018.[25]

The UK is deeply integrated with the EU and its decision to exit from this trading block has no parallel in modern history.[26] Almost one-half of the UK's food is imported: 30% comes from the EU, and another 11% comes from non-EU countries under the terms of trade deals negotiated by the EU. Prices of fruits and vegetables are particularly vulnerable to vagaries of production and supply.[1] In estimates of the economic impact of Brexit on the UK, the least damaging scenarios are those which are closest to the current situation under EU membership (ie, retaining membership of the Single Market and Customs Union), while a 'no-deal' scenario is predicted to be the most damaging.[27]

In setting a comprehensive strategy for the UK to ensure household food security, policymakers must grapple with how to prioritise low food prices, animal welfare, minimum income, health, welfare and social protection.

In anticipation of Brexit in March 2019, we conducted an elicitation of a group of experts in July 2018 on their expectations for possible impacts of two Brexit scenarios on food prices over the 14 months following Brexit (ie, to June 2020). This period was chosen to recognise that there is a period of transition to any new regime and that what we are interested in is what post-Brexit food prices will settle to after the initial volatility. Experts were asked to integrate into their judgements all factors relevant to the changing of food prices at current exchange rates.

## METHODS

A projection of food prices gives a key insight into the effects of unprecedented events on household food security and contingent health consequences. As far as we know, no projections have been published for the impact of Brexit on food prices or CPI which enumerate associated uncertainties formally. That said, one new study has estimated potential impacts of Brexit on the prices of fruits and vegetables and the uncertainties in these using Monte Carlo simulation.[28] Other studies provided only point estimates. We report key findings from the application of a structured expert judgement (SEJ) elicitation to potential food price changes and their uncertainties in the event of Brexit under two scenarios: 'Deal' and 'No-deal'. By Brexit Deal we mean with trading arrangements broadly similar to the present. With Brexit No-deal we mean that such arrangements will be discontinued and individual trade deals would need to be negotiated.

Since SEJ involves the combination of expert judgement, diversity of experts is more important than large numbers. Literature supports 8–15 experts as a viable number in practice; having greater numbers may not significantly impact the findings and would incur extra expense and time. Having fewer than five experts reduces the prospect of providing adequate diversity of views and could weaken the strength of the inferences.[29] We identified potential experts through literature search and scanning web pages of relevant organisations. We sent invitations to 43 individuals whose expertise represented a wide range

within the domain. Of these, only six were able to spend 3 days at the elicitation workshop, so we rescheduled and sent out another tranche of 67 invitations. Again only six could commit to the workshop, so we rescheduled once more, expanded our list of potential experts and issued 81 invitations. Following this iteration, we decided to go ahead with the six external experts and supplement the panel with additional academic colleagues. Two external experts were then late withdrawals from the panel, so we added two more academic volunteers with relevant expertise bring the panel up to 10 specialists in all. Our panel (see the Acknowledgements section) had expertise in food procurement, retail, agriculture, economics, statistics and household food security, and each expert considered potential impacts of the Brexit scenarios on prices separately for each of 10 food categories that are used in the UK CPI. We then used these results to explore a number of scenarios.

## Structured expert judgement

SEJ elicitation provides a formal approach for estimating uncertain quantities according to current professional knowledge and understanding, and has been used in a wide range of applications. To amalgamate our experts' judgements, we used Cooke's Classical Model, a well-established, validated approach.[30–33] Cooke's method uses a mathematical scoring rule basis to evaluate empirical performance-based weights for aggregating individual experts' judgements. This means that when all the individual estimates are combined, the experts who were most informative and most accurate on a set of related calibration questions (to which only the analysts had the answers) contributed most to the final estimates of the questions of interest.[34]

The independent facilitator (WA) devised calibration questions based on historic food price changes. The questions of interest were future food price changes. We began by discussing with the group the wordings and meanings of the questions, to minimise ambiguities or misunderstandings. The experts themselves then clarified and revised several of the target questions.

For both calibration and target questions, we asked the experts to integrate, within their judgements, all the factors and circumstances under which food prices could be driven up or down. Each expert provided their own estimates, confidentially to the facilitator (WA), for the lowest plausible, highest plausible and best estimate for price change for each food category under the specific scenario. We discussed with the experts that we would treat their three judgement values for each item in the subsequent analysis as analogous to a 90% credible interval range, with their best estimate value representing the median of their uncertainty spread. We made clear that a median value need not necessarily be central to the credible interval if, in the expert's judgement, the relevant uncertainty distribution is not symmetric and should exhibit skewness (either to higher or lower values). More

details are given in online supplementary information file S1.

This confidential elicitation procedure encourages participants to express judgements based on their true beliefs, reducing potential sources of bias due to peer and group influences. We also maintained anonymity when presenting to the group the individual judgements and weights derived from the calibration questions.

## Analysis of elicitation

We aggregated mathematically individual uncertainty distributions for each food category to construct food price change probability density functions, using the performance-based weights derived from the expert calibration step in the Cooke's Classical Model.[30]

We processed our experts' responses using the program EXCALIBUR,[35 36] which computes weighted combinations of judgements and produces a synthesis for each food item expressed in terms of the same three quantiles used in the elicitation (ie, 5th, 50th and 95th percentiles). We report in the next section both the overall food price change using the category weights employed by the CPI and those for a healthy basket based on MacMahon and Weld.[37]

## Computing Brexit-related food basket cost changes

In order to estimate price change distributions for different shopping baskets under the two Brexit scenarios we use the Bayes Net (BN) code UNINET.[38] BNs are graphical tools for representing and computing high-dimensional joint uncertainty distributions.[39 40] Our BN for calculating food basket price changes post-Brexit is shown in figure 1.

In postprocessing the elicitation data it was recognised that implicit correlations existed between certain foodstuff prices in the judgements of many of the experts. Ideally, such correlations need to be accounted for in an uncertainty analysis to avoid creating spurious results; here, to mitigate their absence, we adopted approximate correlation values from our knowledge of foodstuff pricing.

The converse assumption, that food prices are independent, introduces the risk of underestimating the joint extent of inter-related changes on distribution tails.

Vegetable and fruit prices are correlated by a mutual dependency on weather conditions. However, both within and between food categories there can be differential effects on different crops: for example, spring floods which rot potatoes and delay grain sowing do no harm to orchards, which suffer if there is an unseasonal frost as the fruit is setting. We set this correlation at about +0.75

Corn and sugar beet are used as feed for cattle, inducing a correlation between meat and grain (bread), and between meat and sugar; we set these correlations each at +0.4. While dairy and beef cattle markets are quite distinct, their prices are linked through a common dependency on feed costs, so we link milk and meat with a correlation +0.4.

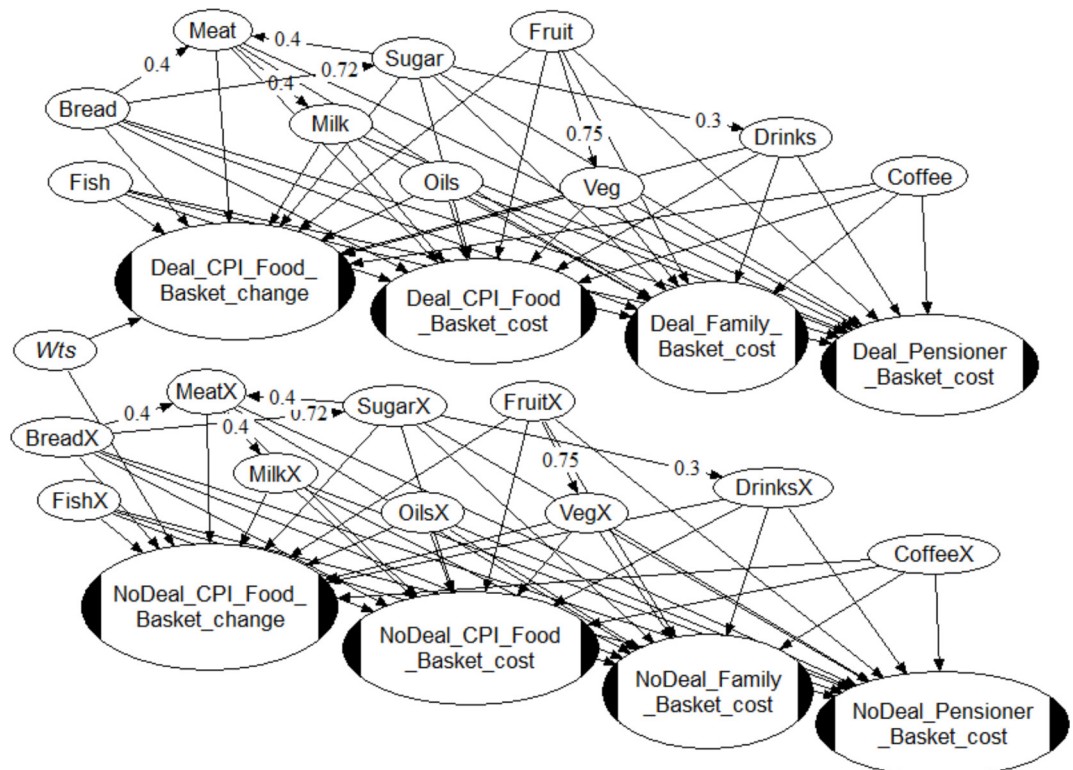

**Figure 1** Bayes Net structure for calculating distributions for food basket price changes (ellipses with black ends) due to elicited judgements on individual foodstuff price movements under Brexit Deal and No-deal scenarios: percentage change in CPI food basket cost; cost change in £ for CPI food basket, and for two household baskets. The information nodes in the upper half of the Bayes Net (Bread, Meat, and so on) comprise uncertainty distributions on price movements per foodstuff for the Brexit Deal scenario; the nodes in the lower half (BreadX, MeatX, and so on) represent uncertainty judgements for foodstuff price movements under a Brexit No-deal scenario. The quantified changes in the basic CPI basket(s) are factored with Office for National Statistics foodstuff weights (node 'Wts'). Numerical distribution statistics for the output nodes are summarised in table 1 (see online supplementary information file S2 for further details). CPI, Consumer Price Index.

Grain (bread) and sugar (beet) can both be feedstock for biofuel and thus are linked via oil price; to account for this and other, secondary joint correlations, we set this correlation to +0.72. In fact, all food prices are linked to oil price through production and transport costs, but these two more strongly.

The 2016 'sugar tax' saw food producers make a substantial switch to artificial sweeteners, smaller serving sizes, and so on, to avoid significant price rises being passed to the consumer. In light of this, we set the correlation between sugar and soft drinks to a relatively low value of +0.3.

## RESULTS

We present estimated projected price changes by June 2020, assuming Brexit on 29 March 2019, using food price change quantiles elicited at food category level (table 1). We use the BN in figure 1 to combine these price changes and associated uncertainties for the food element of the CPI basket, and to estimate monetised equivalents for specific family types based on MacMahon and Weld.[37] We discuss the likely effects on health and the demand for health services. The results of the analysis are in table 1 and are shown graphically in online supplementary figure

A1. Negative values indicate that the experts judge that, under a given scenario, some prices could conceivably go down, as well as up, although with low probabilities.

Using the Office for National Statistics food category weightings, we aggregate their contributions to a CPI basket. We see that, under a Brexit deal, this gives expected median food price rises around +6% (ie, a rise 12 times higher than in 2018) by June 2020, with a plausible, that is, 1-in-20 (5%) likelihood of a drop in prices of about −3% or more and a 1-in-20 chance of a rise of +17% or more. Under Brexit No-deal, the overall median food price escalation is expected to be +23%, with a lower plausible increase of about +1% and upper plausible increase of about +52% (again, each bound has a 1-in-20 chance of being exceeded).

Thus, the foreseeable most likely outcome of Brexit is significant food price rises. This will lead to more household food insecurity and its attendant deterioration in health and increase in demand for health services.

### What-if scenario sensitivity testing

We wished to investigate how sensitive the overall basket food cost results are to a single food type. Currently, the only UK differentials from global prices are beef and poultry, where the EU's production standards are

**Table 1** Aggregated food price change estimates

| | Food category percentage price changes by June 2020 Median (5th, 95th percentiles) | |
|---|---|---|
| CPI category | Brexit Deal | Brexit No-deal |
| Soft drinks and so on | 6 (0, 26) | 8 (0, 47) |
| Coffee, tea and cocoa | 2 (−9, 19) | 4 (−5, 69) |
| Sugar, jam, and so on | 7 (−9, 20) | 19 (−5, 82) |
| Vegetables | 3 (−10, 20) | 9 (−18, 63) |
| Fruit | 5 (−10, 24) | 16 (−8, 51) |
| Oil and fats | 5 (−9, 20) | 18 (−8, 87) |
| Milk, cheese and eggs | 6 (−9, 20) | 23 (−5, 82) |
| Fish | 4 (−9, 19) | 5 (−13, 41) |
| Meat | 6 (−10, 29) | 18 (−11, 80) |
| Bread and cereals | 4 (−9, 19) | 10 (−7, 83) |
| Overall % change Office for National Statistics CPI subfoods, with category weights | Mean +6.4%±6.0 Median +6.1% (−2.7 to +16.9) | Mean +24.0%±15.4 Median +22.5%(+1.49 to +51.7) |
| | **Food basket cost changes by June 2020 (in £s)** | |
| Change in CPI weekly cost relative to 2018 year-end basket total £58.00* | Mean +£3.78±£3.76 Median +£3.53 (−£1.90 to +£10.41) | Mean +£13.97±£9.52 Median +£13.00 (+£0.08 to +£31.02) |
| Change in family of 4 healthy food basket basis weekly cost £93.56† | Mean +£6.30±£6.71 Median +£5.80 (−£3.68 to +£18.17) | Mean +£22.58±£16.14 Median +£20.98 (−£1.07 to +£50.98) |
| Change in single pensioner healthy food basket basis weekly cost £35.44† | Mean +£2.28±£2.56 Median +£2.09 (−£1.51 to +£6.80) | Mean +£8.11±£6.23 Median +£7.55 (−£1.05 to +£18.99) |

'Brexit deal' means a deal similar to the present arrangements will be implemented, so little disruption or additional costs to supply routes. 'Brexit no-deal' means that such arrangements will be discontinued and individual trade deals would need to be negotiated. Numerical values are medians (90% credible intervals).

*Based on Office for National Statistics Table A2 2018 year-end data (March 2018): selected basket subfood category weekly costs; total for the 10 items=£58.00.

†Based on MacMahon and Weld[37] Northern Ireland minimum essential healthy basket subfood category weekly costs at November 2014 Tesco prices. For two adults and two children, one in preschool (aged 2–4) and one in primary school (aged 6–11), total cost for the 10 items=£93.56; for a single pensioner, the corresponding selected items cost=£35.44.

CPI, Consumer Price Index.

higher than the rest of the world.[41] This suggests that constraining price rises for these foods following Brexit might be achieved principally by lowering animal welfare and food hygiene standards, with associated risks to human health.

We set cost of meat to its 5th percentile level and then its 95th percentile level and investigated the effects on the overall food basket cost, with other food types unchanged. The resulting changes to the CPI food basket and to the family basket cost projections reported in table 2 and online supplementary figures A2 and A3.

With meat cost at 5th percentile levels in the elicitation distributions (−10% and −11%, Deal and No-deal, respectively, see table 1), the CPI basket mean change for Brexit Deal becomes −0.1% (compare +6.4%) and the family basket mean cost changes from +£3.78 to −£1.26. Under the Brexit No-deal scenario the CPI basket mean change would be +8.6% (compare +24.0%) and, for the family basket, the cost increase becomes +£5.71 (compare +£13.97).

With meat cost at 95th percentile levels from the elicitation Deal and No-deal distributions (+29% and +80%, respectively), the CPI basket mean change for Brexit Deal increases from +6.4% to +13.5% and the family basket mean cost rises from +£3.78 to +£14.66. Under the Brexit No-deal scenario, the family basket, the mean change would be +44.0% (compare +24.0%), the cost increase becomes +£44.84 (compare +£13.97).

The mean (expected) CPI basket and family basket cost changes would be kept close to zero only if meat prices were, somehow, limited to near their projected 5th percentile levels—and, with them, the price changes of foods correlated with meat were also curbed—and then only under the Brexit Deal scenario. Otherwise, higher meat prices will inevitably amplify basket cost changes. The standard deviations on the means are smaller under these meat price sensitivity tests, and the correlation structure reduces the kurtosis of the distributions.

Although there are some notable differences in the item price compositions of the CPI food basket and

**Table 2** Example impacts on CPI and family food basket costs from analytical conditioning of the Bayes Net to the 5th and 95th percentile projected costs of meat, under Brexit Deal and No-deal scenarios

**Meat price impacts on food basket mean costs: scenario-based Bayes Net analytical conditioning**

| Scenario | CPI basket subfoods cost: mean percentage change* | Family basket: mean cost change* | CPI basket subfoods cost: 95th percentile percentage change* | Family basket: 95th percentile cost change* |
|---|---|---|---|---|
| Meat price → projected 5th percentile cost | | | | |
| Deal (meat cost −10%) | −0.1% (+6.4%) | −£1.26 (+£3.78) | +6.2% (+16.9%) | +£5.01 (+£10.41) |
| No-deal (meat cost −11%) | +8.6% (+24.0%) | +£5.71 (+£13.97) | +24.4% (+51.7%) | +£21.13 (+£31.02) |
| Meat price → projected 95th percentile cost | | | | |
| Deal (meat cost +29%) | +13.5% (+6.4%) | +£14.66 (+£3.78) | +20.6% (+16.9%} | +£21.61 (+£10.41) |
| No-deal (meat cost +80%) | +44.0% (+24.0%) | +£44.84 (+£13.97) | +62.2% (+51.7%) | +£61.56 (+£31.02) |

*Corresponding base model results are shown in brackets.
CPI, Consumer Price Index.

the family 'healthy' food basket (eg, lower spend in UK CPI basket on meat: £12.80 vs £30.18 per week for Northern Ireland), our analysis shows that the overall cost percentage changes in these two representative household food baskets differ little: for Brexit No-deal, both estimates represent about +22% increases in projected mean costs.

We selected meat for this sensitivity analysis as it is the most likely to change in price depending on the details of any trade deals agreed. For other foods or combinations of foods, the opposite may apply; similar sensitivity analyses are possible.

The key finding is that the most likely effect of Brexit on food prices as calculated using the CPI method is a median rise of 6% if there is a deal and 23% if there is no deal. These represent significant additional costs for household budgets and are highly likely to lead to poorer diets with the concomitant effects on diet-related health.

## DISCUSSION
### Principal findings
Food price rises after Brexit are likely to be significant and may be substantial and changes will be felt by the whole population very quickly.[1] In the light of the expected stagnation of household incomes, this is likely to drive more households into food insecurity. Food is a substantial portion of household expenditure, especially in low-income households.[1] MacMahon and Weld[37] found that, after housing and childcare costs, the highest category of household expenditure in Northern Ireland is on a minimum essential 'healthy' food basket. They found food costs to be more expensive in rural areas for all household types and the highest spend was on meat, followed by fruits and vegetables. Those households buying most would, of course, incur the greatest actual spend increases, with concomitant implications for affordability in terms of differing household-related incomes. The well-established links between lower budgets available

for food and lower diet quality lead to the expectation of multiple negative outcomes. Nutrient-dense foods, such as fruits and vegetables, are often more expensive and less available in lower income neighbourhoods when compared with processed foods. Processed foods are generally inexpensive and highly accessible. They are energy dense, high in added fats, sugar, or salt and often considered highly palatable with addictive potential.[21] Increasing numbers of patients with this kind of diet will likely drive increases in the incidence of diet-sensitive chronic diseases in the longer term, such as hypertension, hyperlipidaemia and diabetes.[13–19] The downturn in fruit and vegetable intake has been estimated between 2.5% and 11.4%, dependent on the Brexit trade agreement, with increasing cardiovascular disease mortality.[28] This, in turn, will drive increased demands on the health services. In the short term, this might manifest in reduced control of existing chronic conditions such as diabetes, coeliac disease and hypertension, leading to demand on front-line and general practice services.

### Strengths and weaknesses of the study
One strength of our analysis is that the expert judgement median estimates we obtain by elicitation are consistent with central estimates produced by UK Trade Policy Observatory and by the British Retail Consortium[1] and other modelling studies.[28] However, we add further information for decision support by presenting quantified uncertainties around our estimates. These spreads can be substantial and all exhibit skew in the form of extended, 'heavy' upper tails—that is, larger price increases are more likely than smaller increases (or reductions). Related decision-making that is based only on central (average) estimates and neglects these uncertainties can lead to poor policy selection.[42]

Policymakers benefit from consideration of a defined reasonable worst case for contingency planning, and are proficient at interpreting probabilistic statements, which help clarify underlying assumptions.[43] Moreover, historic

vegetable CPI rises offer a perspective for the elicited credible intervals for CPI vegetable price change in table 1. The largest historic 1-year change in vegetable prices since 1987 was +14.8% (2006–2007) and the largest 2-year jump was +27.1%, for the period 2006–2008. Compared with the Brexit Deal scenario, the record of 2-year rise is greater than the elicited 95th percentile vegetable price index change (ie, +20% by June 2020, 2 years ahead from the elicitation), and falls within the 90% credible interval for Brexit No-deal; the projected 95th percentile change could exceed +63% under this scenario. One weakness of the study was that we had fewer than the optimal number of experts. While the spread of expertise was good, a wider spread of expertise could have improved quantification of price changes, possibly narrowing the uncertainty around the central estimates.

### Strengths and weaknesses in relation to other studies

Although updating is warranted, our likely price change projections exemplify a basis for undertaking a detailed modelling of impacts on health and healthcare provision under the two alternative Brexit scenarios. Most attempts to quantify food prices after Brexit have not been set within the context of health and healthcare provision, nor have they quantified the uncertainty in such estimates; one notable exception is the estimation of the potential impacts of fruit and vegetable price increases on cardiovascular disease.[28] While the experience of food insecurity is largely driven by the cost of food and other essentials relative to incomes, other factors such as self-efficacy, access to credit and other forms of social capital are also significant.[44]

### MEANING OF THE STUDY

The likely effect of Brexit, under either scenario, is a significant rise in food prices. Unless the rising tide of food insecurity is reversed, health costs will continue to rise with implications for health and clinician workloads. Our findings should alert policymakers to the potential for significant increases in food costs under either Brexit scenario, with major impacts likely to follow a No-deal outcome. The expected levels of these increases and, more importantly, the uncertainty spreads on the estimates—all of which are skewed moderately towards higher costs—should inform policies that allow households to afford minimum essential food baskets, meeting acceptable physical, psychological and social needs. One likely corollary to substantial post-Brexit food price rises is even greater consumption of cheaper, less healthy diets, with inevitable impacts on population long-term health trends and demands on the National Health Service. Medical practitioners and healthcare workers are among those who will have to confront the related challenges if food prices rise sharply and substantially after Brexit. Clinicians may need to organise processes to cope with the demand from food insecure patients for referrals to food banks. The management of chronic conditions may deteriorate, increasing the need for intervention. Food insecurity in the present may also increase healthcare demand in the longer term. Policies that support incomes and access to food are the only interventions that have a demonstrable positive effect on food insecurity. It is incumbent on policymakers to ensure that the last line of defence against hunger is robust in order to promote well-being, productivity and healthy ageing.

Current evidence on the efficacy of some of the most widespread current responses to food insecurity, such as food banks, is limited. More research is needed in order to select the most effective strategies for supporting health and well-being. While Brexit did not take place by the due date, and the overall political situation is now even less clear than before, certain aspects of potential impacts on food supplies have become more explicit. For example, the impact of stockpiling on non-perishable food prices can be now ascertained more accurately, as those costs can be quantified. In other respects, however, uncertainties associated with post-Brexit food pricing have increased.

### Unanswered questions and future research

Given almost everything concerning Brexit is in a state of flux at the time of writing, it is our intention to conduct an updating elicitation, once the circumstances surrounding how Brexit will be implemented become clearer. Experts who participated in our 2018 elicitation have expressed a universal interest in revisiting the issues and factors, and in reviewing their judgements of future food prices and the related uncertainties. We will consider increasing membership of the panel of experts to obtain an even wider sample of judgements. One advantage of this repeat elicitation would be the opportunity to investigate how much the quantified uncertainties, reported here, may have changed. When we know the terms of Brexit, an elicitation update will be able to quantify possible adjustments to the present estimates. In addition, we intend to explore in more quantitative detail possible underlying correlations between prices of different foodstuffs. New advances in elicitation techniques for assessing parameter dependencies[45] offer a ratifiable basis for quantifying the properties of such correlations. It seems inevitable that the implications of an eventual Brexit for projected food price forecasts will be fundamental to, and crucial for policy planning in many societal and political areas.

**Acknowledgements** We acknowledge with gratitude the invaluable contributions of the experts who participated in the elicitation workshop, 4–6 July 2018: MJB (University of Warwick, Department of Statistics); Steve Brewer (University of Lincoln, Lincoln Institute for Agri-Food Technology (LIAT), network coordinator for the Internet of Food Things Network Plus); Rosemary Collier (University of Warwick, Life Sciences, Crop Centre); Peter Crosskey (freelance food journalist, formerly of The Grocer); Jonathan Horsfield (retired from Warwickshire County Council; energy expert); Jim Q Smith (University of Warwick, Department of Statistics); Thijs van Rens (University of Warwick, Department of Economics); Rachel Wilkerson (University of Warwick, Department of Statistics); Sophia Wright (University of Warwick, Department of Statistics). Other participating experts from industry preferred to remain anonymous. We thank the reviewers Anthony Laverty, Martin McKee and Abigail Colson for insightful comments which have improved the manuscript.

**Contributors** MJB conceived the study and organised the workshop and experts' participation. WA advised MJB on the elicitation and acted as a neutral independent facilitator for the elicitation and processed the experts' responses. MJB provided information for contextualising the elicitation findings, and both authors jointly wrote the paper.

**Funding** The workshop was funded by the Warwick Global Research Priority for Food. The study is part of work undertaken for EPSRC grant number EP/K007580/1.

**Competing interests** None declared.

**Patient consent for publication** Not required.

**Provenance and peer review** Not commissioned; externally peer reviewed.

**Data availability statement** Data are available upon reasonable request. The first author may be contacted for requests for data or information used in this study.

**ORCID iD**
Martine Jayne Barons http://orcid.org/0000-0003-1483-2943

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
