## [Reviewer comments · BMJ Open]

ARTICLE DETAILS

TITLE (PROVISIONAL)	Anticipated impacts of Brexit scenarios on UK food prices and implications for policies on poverty and health: a structured expert judgement approach
AUTHORS	Barons, Martine; Aspinall, Willy

VERSION 1 – REVIEW

REVIEWER	Anthony Laverty Imperial College London, England
REVIEW RETURNED	22-Jul-2019

GENERAL COMMENTS	Thanks for sending me this paper to review on the very topical issue of the potential impacts of Brexit on food prices. The authors are I think to be commended for having the great foresight to pursue this research at the time and the idea to use expert elicitation is a good one • In the abstract I am not sure that the impact of food prices on physicians through their needing to refer to food aid is the most prominent aspect of the study (although this is the authors choice). Also the authors have not really set a pathway by which increases in food prices will impact management of chronic conditions so this reads perhaps a little oddly• Also the abstract would benefit from the detail on when the expert elicitation took place• I think that the introduction would benefit from some restructuring and having a more explicit focus on brexit. I am not sure that the details around austerity and so on are currently well integrated into the narrative here• The method of expert elicitation is quite new to me but this is well covered I think. The one big issue which did not seem to be covered was how participants were recruited? Looking at the acknowledgements it seems that 6 / 10 of these are from the same institution as the authors. This is not necessarily an issue but I think that there needs to be more clarity on the issue here• The authors do not give any references on the section on how they set correlations etc. between different foods on page 5. Perhaps it is the case that there is not published data on this but then I would like this fact to be much clearer. This section could then be reframed as assumptions which have been made in the absence of good data• The other issue which the authors do not mention is the decision to project prices at 14 months post leaving – I would like to see some form of rationale for this• I also think that detail of sensitivity analyses should be brought up first in the methods. I also found the description of the results here a little unclear and was not sure what the key finding of this section
---

	was  • There is a reference which just says “cite loopstra thesis” • The authors mention that there was a suboptimal number of experts – what would have been the optimal number? And why was it not possible to recruit more people? Also, are we sure that having more experts would have narrowed uncertainty? What if they disagreed with this assessment? • I think that the discussion could do more to present a picture of how these findings fit into the wider literature (some of which if grey) on the impacts of Brexit, which has grown in the last year or so. Also, how does Brexit fit into a larger picture around changes to trade and so on?
--	--

REVIEWER	Martin McKee LSHTM, UK
REVIEW RETURNED	03-Sep-2019

GENERAL COMMENTS	This paper attempts to anticipate some of the consequences of food shortages in two scenarios, Brexit with a deal and without one. As the introduction makes clear, food security is an important determinant of health. Crucially, the United Kingdom is moving towards Brexit in a situation where food insecurity is already widespread, with growing numbers of people dependent on food banks. The authors make a good point that the inflation rate is based on a standardised basket of goods, and I wonder if they might spell out more clearly that, without additional information, it is not possible to assess the extent to which people are shifting from healthy to unhealthy options. This is implicit in what they say that it could be more clear perhaps. In this section, and while realising that there is limited space, where they do note that much food is imported, they might possibly make reference to the risk of further inflation due to the continuing collapse of the value of the pound. There are a few places where I wonder if the language could be tightened up a little. For example, when mentioning the report of the special rapporteur, the point about a single measure of poverty, while interesting, might divert from the key message for this paper that he proposed monitoring food security. The use of structured expert judgement is, as the authors note, a well-established method but for readers who are less familiar with it, it might be useful to have a box setting out how it has been validated. This might only need a few sentences. The inclusion of calibration questions is extremely important, as otherwise one is simply summarising ignorance. However, there is still an issue as to how these panels are selected. While I have no reason to believe that this is the case here, there have been other examples of structured decision-making where there have been serious concerns about the selection of those participating. Consequently, I think more elaboration about how they were chosen would be helpful. I appreciate that some participants wish to remain anonymous but I feel this is a concern. Those that are named do seem to have been drawn disproportionately from Warwick University. You will, no doubt, have a review from an expert statistician. I was pleased to see that they do not treat foodstuffs as independent of each other. However, I would be interested to know how they obtained some negative values for 5th percentiles. I assume this is a function of the methods used in EXCALIBUR but maybe a brief explanation would help? The sensitivity analysis is an important element. One can quibble with the assumptions. For example, while beef from outside the EU
---

	will likely be cheaper (but at the cost of lower welfare standards) there will be additional transport costs. What is important is that they are transparent, as they are here.
--	---

REVIEWER	Abigail Colson University Of Strathclyde, UK
REVIEW RETURNED	02-Dec-2019

GENERAL COMMENTS	This article applies Cooke's Classical Model of structured expert judgement (SEJ) to better understand the possible impact of Brexit on UK food prices. The article is an important and timely contribution to better understand the possible health and well-being impacts of Brexit. Major comments:  1. This is an important application area, however little detail is given on how Cooke's Classical Model has been applied. The paper would be strengthened by more detail, perhaps in supplementary material, about how the experts were recruited, the elicitation questions, the individual expert responses to those questions, and the results of the performance weighting. Important results of the elicitation are omitted in the paper, such as how much agreement or disagreement existed between the experts. The discussion says a weakness of the paper is the number of experts and the spread of expertise; what would ideally have been done, and are there implications of having something less than that? 2. It is not clear in the paper where the correlations used in the UNINET analysis come from. These aren't mentioned as being elicited, so are they based on the author's assumptions or some other source? The correlation between vegetable and fruit prices is 0.75, and the correlation between grain and sugar is 0.72. These differences seem very precise if the correlations are based on authors' assumptions. Is this level of precision warranted? The supplementary material says that the authors do not claim that these correlations are precise and that additional work is warranted in this area, but that should also be stated in the body of the main paper. 3. The discussion says that "other attempts to quantify food prices after Brexit have not been set within the context of health and healthcare provision," implying that this paper does that. Although that context is described in this paper, the paper does not model or estimate the impact on health or healthcare provision. This framing of the paper thus seems to overstate what the paper actually does.
--

	Minor comments: P. 3, line 48 has a typo. It should read “its health consequences.” P. 3, line 49-50 says that the authors are not aware of any attempt to formally enumerate the uncertainty associated Brexit’s impact on food prices. Are there informal attempts, however, that could be discussed? P. 4, lines 4-6 lists the expertise of the experts in the elicitation panel. Did any of the experts have a background in trade law, and were thus familiar with the sort of agreements that would be negotiated in a no-deal scenario and the timelines those agreements would take to implement? Was any background on this provided to the experts for whom this was outside their expertise? P. 5, lines 7-8: “CPI weighted combinations” and “family health basket combinations” should be defined or explained. Table 1: This shaded region of table is a bit confusing to read as written, with both means and medians written in some places but not others. What does the range around the mean indicate? It doesn’t seem to be defined. Table 2: What are the numbers in parentheses in this table? P. 9, lines 27-31: This sentence is confusing as structured. P. 9 line 36: The Loopstra citation is mentioned but missing.
--	---

VERSION 1 – AUTHOR RESPONSE

Reviewer: 1 Reviewer Name: Anthony Laverty Institution and Country: Imperial College London, England Please state any competing interests or state ‘None declared’: None declared Thanks for sending me this paper to review on the very topical issue of the potential impacts of Brexit on food prices. The authors are I think to be commended for	We thank the reviewer for this comment.
---	--

having the great foresight to pursue this research at the time and the idea to use expert elicitation is a good one	
 In the abstract I am not sure that the impact of food prices on physicians through their needing to refer to food aid is the most prominent aspect of the study (although this is the authors choice). Also the authors have not really set a pathway by which increases in food prices will impact management of chronic conditions so this reads perhaps a little oddly 	The conclusions have been re-worded as follows: Conclusions The number of households experiencing food insecurity and the severity of food insecurity are likely to increase because of expected sizeable increases in median food prices after Brexit. Moreover, uncertainty in anticipated food prices is skewed, making higher increases more likely than lower rises. Towards the projected upper limits, these prices would entail severe impacts. Research showing a low food budget leads to increasingly poor diet suggests that demand for health services in both the short and longer term is likely to increase due to the effects of food insecurity on the incidence and management of diet-sensitive chronic conditions.
 Also the abstract would benefit from the detail on when the expert elicitation took place 	Updated the design element of the abstract to include: In July 2018, ten experts estimated the median, 5% and 95% quantiles of changes in price for ten Consumer Price Index food categories under Brexit Deal and No deal to June 2020, assuming Brexit had taken place on 29th March 2019.
 I think that the introduction would benefit from some restructuring and having a more explicit focus on brexit. I am not sure that the details around austerity and so on are currently well integrated into the narrative here 	The text has been re-ordered and augmented, with a summary paragraph to begin the section.
The method of expert elicitation is quite new to me but this is well covered I think. The one big issue which did not seem to be covered was how participants were recruited? Looking at the acknowledgements it seems that 6 / 10 of these are from the same institution as the authors. This is not necessarily an issue but I think that there needs to be more clarity on the issue here	Added (also in response to Reviewer 2, below): Since SEJ involves the aggregation of expert judgements, diversity of experts is more important than large numbers. Literature supports 8–15 experts as a viable number in practice; having greater numbers may not significantly impact the findings and would incur extra expense and time. Fewer than five experts reduces the prospect of capturing an adequate diversity of views and could weaken the strength of inferences(28). We identified potential experts through literature search and scanning webpages of relevant organisations. We sent invitations to 43 individuals whose expertise represented a wide range within the domain. Of these, only six were able to spend three days at the elicitation workshop, so we rescheduled and sent out another tranche of 67 invitations. Again, only six could commit to the workshop, so we rescheduled once more, expanded our list of potential experts and issued 81 invitations.

	We decided to go ahead with the six external experts and to supplement the panel with additional academic colleagues. Two external experts were then late withdrawals from the panel, so we added two more academic volunteers with relevant expertise to bring the panel up to ten specialists in all.
The authors do not give any references on the section on how they set correlations etc. between different foods on page 5. Perhaps it is the case that there is not published data on this but then I would like this fact to be much clearer. This section could then be reframed as assumptions which have been made in the absence of good data	Added: In post-processing the elicitation data it was recognised that implicit correlations existed between certain foodstuff prices in the judgments of many of the experts. Ideally, such correlations need to be accounted for in an uncertainty analysis to avoid creating spurious results; here, to mitigate their absence, we adopted approximate correlation values from our knowledge of foodstuff pricing.
The other issue which the authors do not mention is the decision to project prices at 14 months post leaving – I would like to see some form of rationale for this	Added: This period was chosen to recognise that there is a period of transition to any new regime and that what we are interested in is how post-Brexit food prices will settle to after any initial volatility. Our projection timescale of 14 months covered the anticipated No-deal transition period, and our panel was asked to consider price impacts prior to any agreement(s)
I also think that detail of sensitivity analyses should be brought up first in the methods. I also found the description of the results here a little unclear and was not sure what the key finding of this section was	Added to the methods: We explored the sensitivity of the food basket cost to the price of meat.
There is a reference which just says “cite loopstra thesis”	Citation corrected
The authors mention that there was a suboptimal number of experts – what would have been the optimal number? And why was it not possible to recruit more people? Also, are we sure that having more experts would have narrowed uncertainty? What if they disagreed with this assessment?	Added (as above): Since SEJ involves the aggregation of expert judgements, diversity of experts is more important than large numbers. Literature supports 8–15 experts as a viable number in practice; having greater numbers may not significantly impact the findings and would incur extra expense and time. Fewer than five experts reduces the prospect of capturing an adequate diversity of views and could weaken the strength of inferences(28). We identified potential experts through literature search and scanning webpages of relevant organisations. We sent invitations to 43 individuals whose expertise represented a wide range within the domain. Of these, only six were able to spend three days at the elicitation workshop, so we rescheduled and sent out another tranche of 67 invitations. Again, only six could commit to the workshop, so we rescheduled once more, expanded our list of potential experts and issued 81 invitations. Following this iteration, we decided to go ahead with the six external experts and to supplement the panel with additional academic colleagues. Two external experts were then late withdrawals from the panel, so we added two more academic volunteers with relevant expertise to bring the panel up to ten

	specialists in all.
I think that the discussion could do more to present a picture of how these findings fit into the wider literature (some of which if grey) on the impacts of Brexit, which has grown in the last year or so. Also, how does Brexit fit into a larger picture around changes to trade and so on?	Because the following information was available at the time of the elicitation, we have expanded our discussion in relation to the House of Lords European Union Committee report: “Brexit: food prices and availability”, and Srferidi, Lavery et al 2019: “Impacts of Brexit on fruit and vegetable intake and cardiovascular disease in England: a modelling study”. It would not be valid to relate this elicitation and its findings to subsequent publications which were not available to our experts when they made their judgments; more recent information will be considered in a planned subsequent elicitation.
Reviewer: 2 Reviewer Name: Martin McKee Institution and Country: LSHTM, UK Please state any competing interests or state ‘None declared’: None This paper attempts to anticipate some of the consequences of food shortages in two scenarios, Brexit with a deal and without one.	We thank the reviewer for this comment.
As the introduction makes clear, food security is an important determinant of health. Crucially, the United Kingdom is moving towards Brexit in a situation where food insecurity is already widespread, with growing numbers of people dependent on food banks. The authors make a good point that the inflation rate is based on a standardised basket of goods, and I wonder if they might spell out more clearly that, without additional information, it is not possible to assess the extent to which people are shifting from healthy to unhealthy options. This is implicit in what they say that it could be more clear perhaps. In this section, and while realising that there is limited space, where they do note that much food is imported, they might possibly make reference to the risk of further inflation due to the continuing collapse of the value of the pound.	Added: The CPI on its own cannot indicate if consumers are shifting towards less healthy diets because confounding effects are smoothed out across income ranges. Added: Experts were asked to integrate into their judgements all factors relevant to the changing of food prices at current exchange rates.
There are a few places where I wonder if the language could be tightened up a little. For example, when mentioning the report of the special rapporteur, the point about a single measure of poverty, while interesting, might divert from the key message for this paper that he proposed monitoring food security.	Removed this reference
The use of structured expert judgement is, as the authors note, a well-established method but for readers who are less	See additional supplementary information

familiar with it, it might be useful to have a box setting out how it has been validated. This might only need a few sentences.	
The inclusion of calibration questions is extremely important, as otherwise one is simply summarising ignorance. However, there is still an issue as to how these panels are selected. While I have no reason to believe that this is the case here, there have been other examples of structured decision-making where there have been serious concerns about the selection of those participating. Consequently, I think more elaboration about how they were chosen would be helpful. I appreciate that some participants wish to remain anonymous but I feel this is a concern. Those that are named do seem to have been drawn disproportionately from Warwick University.	We responded to the same point, made by Reviewer 1, above, by adding: Since SEJ involves the aggregation of expert judgements, diversity of experts is more important than large numbers. Literature supports 8–15 experts as a viable number in practice; having greater numbers may not significantly impact the findings and would incur extra expense and time. Fewer than five experts reduces the prospect of capturing an adequate diversity of views and could weaken the strength of inferences (28). We identified potential experts through literature search and scanning webpages of relevant organisations. We sent invitations to 43 individuals whose expertise represented a wide range within the domain. Of these, only six were able to spend three days at the elicitation workshop, so we rescheduled and sent out another tranche of 67 invitations. Again, only six could commit to the workshop, so we rescheduled once more, expanded our list of potential experts and issued 81 invitations. Following this iteration, we decided to go ahead with the six external experts and to supplement the panel with additional academic colleagues. Two external experts were then late withdrawals from the panel, so we added two more academic volunteers with relevant expertise to bring the panel up to ten specialists in all.
You will, no doubt, have a review from an expert statistician. I was pleased to see that they do not treat foodstuffs as independent of each other. However, I would be interested to know how they obtained some negative values for 5th percentiles. I assume this is a function of the methods used in EXCALIBUR but maybe a brief explanation would help?	Added: Negative values indicate that the experts judge that, under a given scenario, some prices could conceivably go down, as well as up, albeit with low probabilities.
The sensitivity analysis is an important element. One can quibble with the assumptions. For example, while beef from outside the EU will likely be cheaper (but at the cost of lower welfare standards) there will be additional transport costs. What is important is that they are transparent, as they are here	We thank the reviewer for this comment.
Reviewer: 3 Reviewer Name: Abigail Colson Institution and Country: University Of Strathclyde, UK Please state any competing interests or state 'None declared': None declared This article applies Cooke's Classical Model of structured expert judgement (SEJ) to better understand the possible impact of Brexit on UK food prices. The article is an important	We thank the reviewer for this comment.

and timely contribution to better understand the possible health and well-being impacts of Brexit.	
Major comments: 1. This is an important application area, however little detail is given on how Cooke's Classical Model has been applied. The paper would be strengthened by more detail, perhaps in supplementary material, about how the experts were recruited, the elicitation questions, the individual expert responses to those questions, and the results of the performance weighting. Important results of the elicitation are omitted in the paper, such as how much agreement or disagreement existed between the experts. The discussion says a weakness of the paper is the number of experts and the spread of expertise; what would ideally have been done, and are there implications of having something less than that?	See additional supplementary information for a record of important elicitation results and the extent of agreement among the experts. Also, we responded to the expert recruitment point, made by Reviewers 1 & 2, above, by adding to the main text: Since SEJ involves the aggregation of expert judgements, diversity of experts is more important than large numbers. Literature supports 8–15 experts as a viable number in practice; having greater numbers may not significantly impact the findings and would incur extra expense and time. Fewer than five experts reduces the prospect of capturing an adequate diversity of views and could weaken the strength of inferences(28). We identified potential experts through literature search and scanning webpages of relevant organisations. We sent invitations to 43 individuals whose expertise represented a wide range within the domain. Of these, only six were able to spend three days at the elicitation workshop, so we rescheduled and sent out another tranche of 67 invitations. Again, only six could commit to the workshop, so we rescheduled once more, expanded our list of potential experts and issued 81 invitations. Following this iteration, we decided to go ahead with the six external experts and to supplement the panel with additional academic colleagues. Two external experts were then late withdrawals from the panel, so wadded two more academic volunteers with relevant expertise to bring the panel up to ten specialists in all.
2. It is not clear in the paper where the correlations used in the UNINET analysis come from. These aren't mentioned as being elicited, so are they based on the author's assumptions or some other source? The correlation between vegetable and fruit prices is 0.75, and the correlation between grain and sugar is 0.72. These differences seem very precise if the correlations are based on authors' assumptions. Is this level of precision warranted? The supplementary material says that the authors do not claim that these correlations are precise and that additional work is warranted in this area, but that should also be stated in the body of the main paper.	Per our response to Reviewer 2, above, we added: In post-processing the elicitation data it was recognised that implicit correlations existed between certain foodstuff prices in the judgments of many of the experts. Ideally, such correlations need to be accounted for in an uncertainty analysis to avoid creating spurious results; here, to mitigate their absence, we adopted approximate correlation values from our knowledge of foodstuff pricing.
3. The discussion says that "other attempts to quantify food prices after Brexit have not been set within the context of health and healthcare provision," implying that this	Reworded: Although updating is warranted, our likely price change projections exemplify a basis for undertaking detailed modelling of impacts on health and healthcare provision

paper does that. Although that context is described in this paper, the paper does not model or estimate the impact on health or healthcare provision. This framing of the paper thus seems to overstate what the paper actually does.	under the two alternative Brexit scenarios. Most attempts to quantify food prices after Brexit have not been set within the context of health and healthcare provision, nor have they quantified the uncertainty in such estimates; one notable exception is the estimation of the potential impact of fruit and vegetable price increases on cardiovascular disease (40).
Minor comments: P. 3, line 48 has a typo. It should read “its health consequences.”	Thank you
P. 3, line 49-50 says that the authors are not aware of any attempt to formally enumerate the uncertainty associated Brexit’s impact on food prices. Are there informal attempts, however, that could be discussed?	This said, one new study has estimated potential impacts of Brexit on the prices of fruits and vegetables, and the uncertainties in these using, Monte Carlo simulation, and on cardiovascular disease rates (28). Other studies provided only point estimates.
P. 4, lines 4-6 lists the expertise of the experts in the elicitation panel. Did any of the experts have a background in trade law, and were thus familiar with the sort of agreements that would be negotiated in a no-deal scenario and the timelines those agreements would take to implement? Was any background on this provided to the experts for whom this was outside their expertise?	This period was chosen to recognise that there is a period of transition to any new regime and that what we are interested in is how post-Brexit food prices will settle to after any initial volatility Our projection timescale of 14 months covered the anticipated No-deal transition period, and our panel was asked to consider price impacts prior to any agreement(s).
P. 5, lines 7-8: “CPI weighted combinations” and “family health basket combinations” should be defined or explained.	Reworded We report in the next section both the overall food price changes using the category weights employed by the CPI and those for a ‘healthy basket’, based on McMahon and Weld (2015)(36).
Table 1: This shaded region of table is a bit confusing to read as written, with both means and medians written in some places but not others. What does the range around the mean indicate? It doesn’t seem to be defined.	Updated to explicitly label means and medians.
Table 2: What are the numbers in parentheses in this table?	(* Corresponding base model results are shown in brackets.)
P. 9, lines 27-31: This sentence is confusing as structured.	Reworded
P. 9 line 36: The Loopstra citation is mentioned but missing	Added.
Please include figure 1 caption at the end of your main manuscript.	Inserted after references: Figure 1 Bayes Net structure for calculating distributions for food basket price changes (ellipses with black ends) due to elicited judgments on individual foodstuff price movements under Brexit Deal and No Deal scenarios: percentage change in CPI Food Basket cost; cost change in £ for CPI Food Basket, and for two household baskets. The information nodes in the upper half of the BBN (Bread; Meat .. etc) comprise uncertainty distributions on price movements per

	foodstuff for the Brexit Deal scenario; the nodes in the lower half (BreadX; MeatX .. etc) represent uncertainty judgments for foodstuff price movements under a Brexit No Deal scenario. The quantified changes in the basic CPI Basket(s) are factored with ONS foodstuff weights (node "Wts"). Numerical distribution statistics for the output nodes are summarised on Table 1. (See Supplementary Information for further details).
--	--

VERSION 2 – REVIEW

REVIEWER	Anthony Laverty Imperial College London
REVIEW RETURNED	06-Jan-2020

GENERAL COMMENTS	I thank the authors for attending to the comments and think this paper worthy of publication.
---

REVIEWER	Martin McKee London School of Hygiene & Tropical Medicine, UK
REVIEW RETURNED	02-Jan-2020

GENERAL COMMENTS	My previous comments have been addressed satisfactorily.
--

REVIEWER	Abigail Colson University of Strathclyde, UK
REVIEW RETURNED	19-Jan-2020

GENERAL COMMENTS	Thank you for the additional information provided in the supplementary materials. I have no further comments.
---